# Sensors on the Internet of Things Systems for Urban Disaster Management: A Systematic Literature Review

**DOI:** 10.3390/s23177475

**Published:** 2023-08-28

**Authors:** Fan Zeng, Chuan Pang, Huajun Tang

**Affiliations:** School of Business, Macau University of Science and Technology, Taipa, Macao 999078, China; fzeng@must.edu.mo (F.Z.); cpang@must.edu.mo (C.P.)

**Keywords:** sensors, Internet of Things, urban disaster management, flood, earthquake, landslide, search and rescue

## Abstract

The occurrence of disasters has the potential to impede the progress of sustainable urban development. For instance, it has the potential to result in significant human casualties and substantial economic repercussions. Sustainable cities, as outlined in the United Nations Sustainable Development Goal 12, prioritize the objective of disaster risk reduction. According to the Gesi Smarter 2030, the Internet of Things (IoT) assumes a pivotal role in the context of smart cities, particularly in domains including smart grids, smart waste management, and smart transportation. IoT has emerged as a crucial facilitator for the management of disasters, contributing to the development of cities that are both resilient and sustainable. This systematic literature analysis seeks to demonstrate the sensors utilized in IoT for the purpose of urban catastrophe management. The review encompasses both the pre-disaster and post-disaster stages, drawing from a total of 72 articles. During each stage, we presented the characteristics of sensors employed in IoT. Additionally, we engaged in a discourse regarding the various communication technologies and protocols that can be utilized for the purpose of transmitting the data obtained from sensors. Furthermore, we have demonstrated the methodology for analyzing and implementing the data within the application layer of IoT. In conclusion, this study addresses the existing research deficiencies within the literature and presents potential avenues for future exploration in the realm of IoT-enabled urban catastrophe management, drawing upon the findings of the evaluated publications.

## 1. Introduction

Cities serve as the primary hubs for economic activities, social interactions, cultural expressions, and overall human existence [1]. It is anticipated that by the year 2050, approximately 86% of affluent nations will have undergone urbanization, while around 64% of developing nations will have experienced the same phenomenon [2,3]. At present, the global urban population stands at approximately 4.27 billion individuals, constituting approximately 55% of the total global population [1,4]. It is anticipated that almost 70% of the global population will undergo urbanization and relocate to urban areas by the year 2050 [4]. This significant shift will probably result in a corresponding expansion of the world’s metropolitan regions, encompassing an estimated additional land area of 1.2 million square kilometers [4].

Cities often have larger population densities, making them more vulnerable to many sorts of disasters. As a result, cities have major impacts as a result of these disasters [5]. Disasters possess the capacity to cause harm to human lives and give rise to unfavorable economic and environmental outcomes [6,7]. From 2001 to 2020, there was an annual occurrence of big and medium-sized disasters ranging from 350 to 500 [8]. Furthermore, it is important to acknowledge that a greater population density results in a heightened demand for rescue services, therefore requiring more sophisticated strategies for catastrophe management and the deployment of disaster relief efforts [9]. The lack of effective communication between public rescue and safety groups, rescue teams, first responders, and persons who are trapped worsens the situation [10]. Furthermore, it is important to acknowledge that disasters possess the capacity to inflict substantial harm against essential infrastructure systems, encompassing, but not limited to, electrical grids, water distribution networks, transportation networks, and communication systems [5]. Disasters have the capacity to disrupt economic activities and yield significant economic losses. Between the years 2008 and 2018, an extensive examination indicates that a cumulative count of 3751 occurrences of natural catastrophes took place, including a diverse range of phenomena, including earthquakes, floods, and tsunamis. The occurrence of these catastrophic catastrophes led to a significant economic downturn, resulting in a total financial loss of $1.658 billion [9]. Urban disasters have the potential to yield substantial environmental ramifications, encompassing the release of pollutants and the handling of waste disposal. In light of the considerable repercussions that catastrophes have on urban environments, leading to enormous losses, it is crucial to improve the management of urban disasters. The achievement of the United Nations Sustainable Development Goal (UNSDG) 12 entails the need to diminish the probability of catastrophic events and enhance the overall resilience of urban regions to withstand and recuperate from such occurrences by the year 2030 [11]. The successful attainment of the UNSDGs relies on the efficient execution of disaster management policies within metropolitan regions [12].

The concept of disaster management entails the systematic coordination and administration of various endeavors during all phases of a disaster, including but not limited to mitigation, relief, response, and recovery [9]. The primary objectives of disaster management encompass the initiation of timely alerts, the acquisition of real-time data, the precise assessment of damages, the prompt identification of evacuation pathways, and the efficient administration of emergency provisions [9]. The conventional methods of disaster management are becoming outdated due to their inability to effectively gather data from various sources in real-time and process and evaluate vast quantities of catastrophe-related information in real-time [9].

IoT enables the collection and analysis of real-time data, presenting opportunities for addressing catastrophe management in urban areas [11,13]. IoT can be described as a framework that facilitates inter-device communication via the Internet [9]. The promise of technology to facilitate complex decision support systems is evident through its ability to deliver services in a more accurate, organized, and intelligent manner [13]. IoT has significantly enhanced the capacity for analyzing catastrophe risks, namely in the areas of floods and earthquakes. This advancement has facilitated the development of more effective disaster response plans and risk management policies [11]. Numerous instances exist wherein the IoT is employed for the purpose of regular surveillance of natural occurrences, transmission of alert alerts, and provision of timely information to disaster management authorities [11]. In 2020, 23 out of 195 UN countries had effective disaster early warning systems, which successfully protected 93.63% of the population at risk from natural disasters in those countries (https://sendaimonitor.undrr.org/analytics/global-target/16/8 (accessed on 19 May 2023)). Flood warning systems can reduce flood losses by 35% annually (https://documents1.worldbank.org/curated/pt/609951468330279598/pdf/693580ESW0P1230aster0Risk0Reduction.pdf (accessed on 19 May 2023)). Early warning systems on the West Coast of the United States enhance population preparedness for the disaster, reducing the risk of injuries by 50% [14]. The utilization of IoT technologies facilitates the acquisition of data, enabling communities to receive periodic updates and implement proactive steps in response to imminent disasters [11]. The IoT technology plays an important role in rescue actions since it can provide instantaneous updates of information [13]. The ability to make effective and precise decisions in a timely manner is crucial during relief operations due to the needs and dynamic nature of the environment [13]. The first 72 h after a disaster (i.e., the golden rescue time) are crucial for search and rescue, as the probability of finding survivors sharply decreases after this period [15]. The implementation of IoT technology has the potential to enhance the effectiveness of search and rescue operations within a designated time frame of 72 h [15]. Hence, IoT has the capability to offer real-time monitoring, timely alerts, post-disaster response, and assistance in rescue operations, thereby assuming a significant role in urban catastrophe management. Furthermore, the utilization of IoT devices has become increasingly prevalent as a cost-effective and straightforward approach to monitoring various systems [16]. Our review aims to offer managers a comprehensive set of recommendations for the effective implementation of IoT technologies in the context of urban catastrophe management.

The architecture of IoT primarily has three layers, namely the perception layer (sometimes referred to as the sensor layer), the network layer, and the application layer [17]. Our review commences by focusing on sensors, which serve as the foundational component of the IoT framework. In the realm of IoT devices, sensors play a pivotal role in the collection and aggregation of data [18]. Sensors have the capability to be deployed in diverse environments, including riverbeds and soil. The sensors have the capability to gather and transmit data in real-time on a continuous and automated basis. Sensors are vital link between the physical and digital realms, assuming a pivotal function within the IoT framework. Subsequently, the data would be conveyed to the application layers for the purpose of data analysis and support applications, utilizing diverse communication technologies and protocols. Then, the data would be transmitted to application layers for data analysis and handle applications through various communication technologies and protocols, facilitated by gateways in the network layer [17]. The utilization of sensor-generated data inside IoT systems can facilitate data analysis and decision-making processes in the field of disaster management, provided that the data is successfully uploaded to the application layers [19]. Hence, the primary objective of this study is to address the research inquiries pertaining to both the pre-disaster and post-disaster stages.

What kinds of sensors are used to collect data? And what kinds of data are focused on the pre-disaster stage and post-disaster stages, respectively?

We aim to present a comprehensive analysis of the sensors employed in many catastrophe scenarios, with a particular focus on investigating the feasibility of developing universal sensors capable of addressing multiple types of disasters during the pre-disaster phase. During the stage following a disaster, it is imperative to deliberate on the appropriate data to be gathered for the purpose of post-disaster management.

2.What kinds of communication technologies and protocols are used to transmit the data from sensors?

We aim to examine the communication technologies and protocols employed during the pre-disaster stage, with the objective of identifying the prevailing and effective communication technologies and protocols. Additionally, we will examine the emergency communication technologies and protocols implemented during the post-disaster phase. The communication technologies and protocols utilized during the post-disaster phase differ from those employed in the pre-disaster phase due to the potential destruction of the communication infrastructure established prior to the occurrence of the disaster.

3.What methods were used to analyze sensor data?

The enhancement of machine learning algorithms has the potential to reduce expenses associated with sensors and facilitate expedited disaster alert systems. Furthermore, the utilization of visual algorithms facilitates the detection of various types of disasters, such as floods and earthquakes, via cameras. Therefore, it is imperative to conduct an investigation into the processes of data analysis.

4.What are the differences between the IoT technologies used in the pre-disaster and post-disaster stages?

Prior research has shown a greater emphasis on the utilization of sensors in the IoT for disaster management in the pre-disaster phase as opposed to the post-disaster phase. Esposito et al. [14] conducted a comprehensive review of early warning systems for natural catastrophes in the pre-disaster stage, specifically focusing on IoT. Ahmed et al. [20] critically examine the utilization of affordable sensors in the monitoring of climate-related disasters in coastal regions. Nonetheless, it is imperative for disaster management to encompass many activities during both the pre-disaster stage, such as disaster identification and prevention, as well as the post-disaster stage, including evacuation, search and rescue operations, and rehabilitation efforts [21,22]. Disasters can manifest abruptly or unexpectedly [6]. The likelihood of locating individuals who have survived a catastrophic event significantly decreases subsequent to the first 72 h period [15]. Efficient disaster response systems are crucial in order to mitigate human suffering and mortality rates [6]. Hence, we not only address the sensors employed in IoT systems during the pre-disaster stage but also emphasize the sensors utilized in IoT systems during the post-disaster stage. Additionally, we conduct a comparative analysis of IoT systems throughout the pre-disaster and post-disaster stages. In this study, our objective is to identify the prevailing technologies, such as sensors, communication technologies and protocols, and data analysis methodologies, that are utilized in IoT-based disaster management throughout both the pre-disaster and post-disaster stages.

The rest of the article discusses the methodology, the IoT system in the pre-disaster stage, the IoT system in the post-disaster stage, and the difference between the IoT technologies used in the pre-disaster and post-disaster stages. Section 2 introduces the methodology (i.e., systematic literature review) used in this study. Section 4 introduces the IoT system in the pre-disaster stage, while Section 5 introduces it in the post-disaster stage, which answers research questions 1–3. Section 6 compares the differences between the IoT technologies used in the pre-disaster and post-disaster stages, which answers research question 4. Section 7 concludes the findings of this study, indicates the limitations, and shows the future research directions in this research area.

## 2. Methodology

Systematic Literature Review (SLR) is a way to synthesize research findings in a systematic, transparent, and repeatable manner for identifying and critically evaluating relevant research to answer specific research questions or hypotheses [23]. Previous studies widely used the SLR approach to organize literature and perform a thorough literature review in the areas of sensors in the IoT [19,20]. Regarding the SLR process, we should first collect the target literature, then identify the literature that meets the pre-specified inclusion criteria, and finally provide solid findings.

To search the target publications, we combined the keywords into three parts: (1) the IoT keywords, (2) the keywords related to “urban”, and (3) the keywords related to natural disasters. Firstly, we target publications related to IoT through the keywords “Internet of Things”, “wireless sensor networks”, and “Internet of Everything” and their abbreviations [14,24]. Further, we scoped the publications in the scenarios of the city through the keywords. “urban”, “city”, and “cities” [25,26,27,28]. Finally, we scoped the publications related to disaster management through the keywords “disaster”, “natural hazard”, “flood”, “landslide”, “earthquake”, “storm”, “hurricane”, “wildfire”, “tornadoes”, “cyclones”, “drought”, “tsunami”, “typhoon”, “avalanche”, “heatwave”, “volcan*”, and “gully erosion” [8,29,30,31,32,33,34,35,36]. Therefore, our final search string was “TS = (“internet of thing*” OR “IoT” OR “IoTs” OR “wireless sensor network*” OR “WSN” OR “WSNs” OR “Internet of Everything” OR “IoE”) AND TS = (urban OR city OR cities) AND TS = (disaster* OR “natural hazard*” OR flood* OR landslide* OR earthquake* OR storm* OR hurricane* OR wildfire* OR tornado* OR cyclone* OR drought OR tsunami* OR typhoon* OR avalanche* OR heatwave* OR volcan* OR “gully erosion*”)”. We used quotation marks to search the whole specific phrase, such as “Internet of Things”. But the quotation marks prevented WoS from searching US and UK spelling variations automatically (http://webofscience.help.clarivate.com.libezproxy.must.edu.mo/en-us/Content/spelling-variations.html (accessed on 19 May 2023)). Thus, quotation marks are not necessary for a single word, such as “urban”. We added an asterisk to prevent variations of words from being missed (http://webofscience.help.clarivate.com.libezproxy.must.edu.mo/en-us/Content/search-operators.html#Search (accessed on 19 May 2023)). It is worth noting that the asterisk is not available to search US and UK spelling variations (http://webofscience.help.clarivate.com.libezproxy.must.edu.mo/en-us/Content/spelling-variations.html (accessed on 19 May 2023)).

We ran the search string on the Web of Science, which is one of the significant bibliographic databases [35]. We selected Web of Science since it allowed us to select a large number of highly credible publications with impact factors [35]. We only considered English publications [35]. The types of publications include research articles, conference papers, and review papers [8]. Finally, we gathered 502 publications.

We excluded irrelevant publications through two stages: screening the title and abstract of the publications and a full review of the publications. After screening the titles and abstracts of the publications, we excluded 400 irrelevant publications. After fully reviewing the publications, we excluded 30 irrelevant publications. Finally, we constructed a database with 72 publications for a systematic literature review. Among 72 publications, 47 focus on the pre-disaster stage, while 20 focus on the post-disaster stage. Five publications both mention IoT systems used in the pre-disaster and post-disaster stages.

## 3. Descriptive Analysis

Figure 1 shows the distribution of sample publications by year. Academic attention was directed towards this particular research domain in the year 2007. However, there was a limited number of papers that concentrated on this specific subject field throughout the subsequent decade. The quantity of publications reached its highest point in 2021, with a total of 17 publications, but experienced a significant decline to 7 publications annually in 2022. The dataset of articles for the year 2023 is still incomplete, as it only includes items downloaded up until May 2023.

Table 1 shows the top six distributions of publications by sources. Among 72 publications, most were published in the journal *Sensors* (five publications). Two publications were published at the 2019 5th IEEE International Smart Cities Conference, Applied Sciences, Earth Science Informatics, IEEE Access, and Materials Today-Proceedings, respectively.

## 4. IoT Systems in the Pre-Disaster Stage

IoTs have various advantages, such as low cost, low energy consumption, access to harsh environments, and simple installation [37]. More importantly, the sensors used in IoT systems can adapt to changes in the environment and collect real-time, high-precision environmental data [37]. Therefore, IoT systems are useful tools for monitoring natural environments and disaster management [37].

### 4.1. Sensors

Sensors used in the pre-disaster stage mainly collect environmental data. Accelerometers are usually used to detect earthquakes [38,39]. Except for accelerometers, more sensors are used to detect landslides, such as inertial sensors, bar extensometers, and borehole inclinometers. More publications focus on flood monitoring. Floods may occur more frequently in cities. One reason is that urban drainage systems often become saturated due to prolonged and intense rainfall [40]. Regarding flood monitoring systems, scholars use more sensors, such as rain gauges [40,41], water level sensors [40,42,43,44,45,46,47], water pressure sensors [41,45,47], cameras [41,48,49], soil moisture sensors [42], weather sensors [42], drones with drones [42], water presence sensors [44,50], temperature sensors [50], and a triaxial accelerometer [50]. The sensors are usually powered by solar batteries [41,50]. To save the energy cost of the sensors, Biabani et al. [51] introduced a model with a harmony search algorithm and improved hybrid Particle Swarm Optimization to select cluster heads. Based on Particle Swarm Optimization, they developed a multi-hop routing system with enhanced tree encoding and a modified data packet format. The single computer boards are the Raspberry Pi [43,48] and Arduino [43,44]. We will further discuss the sensors used for different disaster types in the following content.

#### 4.1.1. Earthquake

We commonly use accelerometers to detect earthquakes. Accelerometers include triaxial accelerometers and dual-axis accelerometers. Regarding triaxial accelerometers, we can choose the accelerometer ADXL362 [39] for low-power use, while we can choose the accelerometer EPSON M−A351AU [39] and the accelerometer LSM9DSO [16] for high-precision use. Regarding dual-axis accelerometers, we can choose the accelerometer ADXL203, which is low-power and high-precision [52]. Also, we can choose triaxial accelerometers, such as L1S3DSH sensors (manufactured by STMicroelectronics) and EpiSensors [53]. The L1S3DSH sensor is ultra-low-power and high-performance [53]. The accelerometers are placed on the object being detected (e.g., buildings or bridges). The single computer boards include Raspberry [38], CC2420 DBK [52], and Sparrow v4 [16]. The microprocessors include the ATmega128L [52], ATmega128RFA1 [16], and ARM processor [53]. Sensors should be equipped with antennas to enable data transmission over long distances [39]. The sensors should be energy-saving [39]. Regarding batteries, we can choose d-cell batteries [39] and CR2032-3V lithium-ion batteries [16]. The sensors should sleep when they do not need to collect data [39]. In addition, the sensors should be low-cost [39]. For example, each sensor in Siringoringo et al.’s [39] earthquake detection system costs 2300 USD.

In addition, some scholars may use other sensors for earthquake detection. For example, Castelli et al. [54] also combined triaxial velocimeters and ultrasonic measurements to build the earthquake early warning system. Tudose et al. [16] combined an LSM9DSO-16 bit high-resolution triaxial accelerometer with an SI7020 humidity and temperature sensor, a three-axis gyroscope, a triaxial magnetometer with embedded FIFO, an SI1145- infrared proximity detection, a high-precision altimeter, a UV and ambient light sensor, and a MPL3115A2-pressure and temperature sensor.

#### 4.1.2. Landslides

People detect landslides with more sensors, such as inertial sensors [55], accelerometers [55], bar extensometers [56,57], borehole inclinometers [56], rainfall sensors (e.g., rain gauge [58]), and displacement meters [58]. The models of inertial sensors include the IMU6050 [55]. The models of accelerometers include LIS3331LDH [55]. The models of microprocessors include the ESP32 [55,59]. The models of single computer boards include the Waspmote PRO board [55]. Usually, the sensors can store data locally on SD cards [55]. Batteries [55,56,57] and solar [56] are the major power sources. To improve energy efficiency, Zhang [60] designed a wavelet-based sampling process for landslide sensors. This process allowed the sensors to reduce data gathering while maintaining performance and system reliability, which allows the battery to run continuously for 3–5 months without recharging during the monsoon period. Wang et al. [57] suggested using the WorkStop recycling control mode in the batteries.

#### 4.1.3. Floods

People develop flood detection systems based on more considerations such as rainfall [40,41,42,61,62,63,64], water level [40,43,44,46,47,62,63,65,66,67,68], water pressure [41,44,45,47,50,69], soil moisture [42], solar radiation [42], vapor pressure [42], relative humidity [42], humidity [42], temperature [42,61,62,70], air pressure [42,62], wind speed [42,61], wind gust [42], wind direction [42], tilt [42], lighting [42], lighting average distance [42], the flow velocity [42,62]. Moreover, Ragnoli et al. [50] also used GSM to detect locations in their flood monitoring systems.

We usually collect rainfall data using rain gauges [40,41], such as double-tipping buckets [40]. We usually measure water level with water level sensors, such as radar level sensors [40,65], ultrasonic sensors [42,47,65,68,70], and force-sensitive resistors [47]. Ultrasonic sensors include MaxBotix MB7066 [70], HC-SR04 [68], and so on. A method to measure water level is to measure water pressure and convert that data into water level [45,64,69]. Ragnoli et al. [50] detected water with electrical resistance. To be specific, they used fork-shaped probes made of conductive and corrosion-resistant metal with 10 cm long and 1.5 cm spaced terminals [50]. The resistance would drop when the terminal came into contact with water [50]. Regarding fault tolerance, they added a triaxial accelerometer (i.e., ADXL345) to use when the water sensors were damaged [50]. Malik et al. [44] combined the Adafruit SHT31-D temperature and humidity sensor with dual ultrasonic sensors to monitor water levels. Another sensor type aims to detect the presence of water. Malik et al. [44] combined a waterproof temperature sensor with dual water presence sensors to detect water presence. Mendoza-Cano, Aquino-Santos, Lopez-de la Cruz, Edwards, Khouakhi, Pattison, Rangel-Licea, Castellanos, Martinez-Preciado, Rincon-Avalos, Lepper, Gutierrez-Gomez, Uribe-Ramos, Ibarreche and Perez [42] combined Drifters and river drones to measure parameters such as river flow velocity and water temperature during the flooding events. Drifters were the main measuring tool, while RiverDrone aimed to locate the Drifters. Mousa, Oudat, Claudel and Ieee [70] suggested measuring temperature through passive infrared sensors (e.g., Melexis MLX90 614).

Cameras are also effective tools for flood detection [41,48,49,63,69,71,72,73]. Regarding the use of cameras, Castro et al. [72] suggested using no infrared cameras, while Castro et al. [72] and Garcia et al. [48] suggested using cameras with water level markers. Regarding water level markers, Castro et al. [72] suggested placing highly visible reflective tapes on surfaces visible to cameras ranging from 0 to 1.5 m. Each tape was spaced out with other tapes, which allowed us to obtain a better approximation of the severity of the water level. This method could improve accuracy because it was not affected by the temperature and humidity of the air or the objects that could absorb wave sounds. In addition, Garcia et al. [48] suggested putting visible marks on the image captured by the camera to detect the flood-severity level.

The types of power supplies are numerous. For example, we can place sensors on electricity poles to absorb power [40,47]. Solar and/or batteries are the common power supply [41,44,50,70], such as Seeed Studio solar cell batteries with TP4056 charge regulators [50] and solar-powered Lithium Iron Phosphate batteries [70]. The models of microprocessors include the Raspberry Pi 4 Model B [48], Arduino Uno [43], Raspberry Pi 3 Model B+ [43], Analog to Digital Converter [50], Intel PXA271 XScale [71], and ESP32 [67]. The models of single computer boards include Arduino [44,68], NodeMCU [47], TelosB [74], Raspberry Pi [72], ARM Cortex M4 [70], and Arduino DUE [67]. We can also consider SD cards for local data storage [70].

#### 4.1.4. Others

Park and Baek [75] introduced the detection of heatwaves and cold waves. Alhamidi et al. [76] presented an IoT-based tsunami monitoring system. Aljohani and Alenazi [77] introduced a storm detection system. We can use meteorological sensors to detect heatwaves and cold waves by monitoring parameters such as temperature, relative humidity, noise, illumination, ultraviolet, vibration, PM10, PM2.5, wind speed, wind direction, CO, NO_2_, SO_2_, NH_3_, H_2_S, and O_3_ [75]. Alhamidi, Pakpahan, Simanjuntak and Iop [76] used the ADXL335 accelerometer to read vibrations in the seafloor crust because tsunamis are caused by vibrations and faults in the seafloor crust. They also connected sensors to flare-marking buoys to provide information to the nearest disaster mitigation center. They used the Arduino Uno as a single computer board. Regarding storm detection systems, Aljohani and Alenazi [77] suggested using weather sensorsincluding humidity and lightningsensors. For forest fires detection, Viegas [78] used sensors to collect data such as temperature, humidity, gas concentrations, rain rate, wind direction, and wind speed. They also use cameras, including Pan-Tilt-Zoom and Fixed Cameras, and Unmanned Air Vehicles equipped with cameras. To detect typhoons, Wang et al. [79] suggested using meteorological satellites as sensors to obtain high-resolution remote sensing image data to recognize typhoon clouds and locate the typhoon center.

### 4.2. Communication Technologies and Protocols

People usually transmit the data from sensors to servers through Bluetooth [38], Ethernet [38,42,43], Wi-Fi [38,43,55,56,59,68,72], and cellular communication technology [42,57,59,71,80], Radio Frequency [42], and radio [44]. Cellular communication technologies include GSM [57], GPRS [57,71], and 3G [42]. The communication protocols include Choco protocol [39], UDP [41], IPv6 with LoWPAN [41], IEEE 802.15.4 [16,41,52], Message Queuing Telemetry Transport (MQTT) [42,43,47,59,67,81], concurrent multi-path transfer protocol [49,82], LoRaWAN [50,62,80], TCP/IP Internet protocol [50,83], Hyper Text Transfer Protocol (HTTP) [61], Zigbee [16,45,71], LRWiFi [59], Cat-M1 [59], CoAP [59], XBee [70]. The data is usually transmitted in JSON format [42,50,61]. Regarding data storage, people may use local data storage (e.g., SD memory cards [55]) and cloud storage (e.g., MongoDB [48], Dynamo [43]). Miao and Yuan [58] used the SQL Server 2008 database software. Malik et al. [44] store the data in Oracle’s MySQL and host the database on an Ubuntu Server. Drones can relay data from sensors to base stations, thereby effectively achieving large-scale data transmission [18]. Drones can cooperate with drifters to collect river velocity data. In this combination, drones aim to locate the drifters and transmit the data from the drifters to a server [42].

The connection solutions for earthquakes include Bluetooth [38], Ethernet [38], Wi-Fi [38], Choco protocol [39], IEEE 802.15.4 [16,52], MQTT [81], and Zigbee [16]. Regarding the connectivity solutions for landslides, we can use Wi-Fi [55,56], cellular communication technology [57,59], MQTT [59], LRWiFi [59], Cat-M1 [59], and CoAP [59]. The connection solutions for floods include UDP [41], IPv6 with LoWPAN [41], IEEE 802.15.4 [41], MQTT [42,43], Ethernet [42,43], cellular communication technology [42,71,80], Wi-Fi [43,68,72], concurrent multi-path transfer protocol [49], LoRaWAN [50,62,80], TCP/IP [50,83], radio [44], HTTP [61], ZigBee [45,71], 6LoWPAN [74], and XBee [42,70]. Concurrent transfer can achieve higher throughput [49,82], accelerate transmission [49,82], reduce packet loss [49,82], and save energy [39]. Choco protocol [39] and 6LoWPAN [74] can save energy. In addition, Luo et al. [84] proposed the “MWAC model” for sensor networks to save power and transmit information over long distances (p. 49).

Usually, we convert information between different protocols using different systems and intermediate devices. For instance, Ragnoli et al. [50] and Gomes et al. [83] both transmitted the data for sensors to the server via TCP/IP while transmitting the data from the server to user applications through HTTP. However, Ferraz et al. [61] built the servers to connect sensors and human clients based on the HTTP protocol without the use of other protocols.

### 4.3. Analysis and Applications of Sensor Data

Firstly, we tend to emphasize that data pre-processing is important to improve the efficiency of data analysis. Some missing sensor data were recorded as zero because of the irregular data transmission and the irregular observation time. We could not distinguish this missing data from an observed zero value. To solve this problem, Park and Baek [75] suggested some quality management for sensor networks, such as data pre-processing (time allocation and filling short gaps in missing data), physical limit check, climate range check, internal consistency check, persistence check, step check, spatial consistency check, spatial outlier check, and data reconstruction using spatial and temporal gap-filling. In addition, Wang and Abdelrahman [62] suggested using a divide-and-conquer approach to process the high-dimensional data inputs from sensors. For example, we can group the sensors by their physical locations and customize the model to process each sensor.

Since sensors have limited resources, another way to improve the efficiency of data analysis is to combine fog computing and cloud computing [41,85]. To be specific, sensors send the data to the fog periodically [85]. After the fog pre-processes the data, it will be transmitted to the cloud [85]. Therefore, fog computing is mainly responsible for concentrating, distributing, caching, and analyzing the data, detecting abnormalities, analyzing the data on a smaller scale, sending notifications and feedback, and forwarding summarized data to the cloud periodically [41,85]. Fog computing can reduce the latency of the service, respond to any emergency change immediately, and reduce the burden on the cloud [41,85]. Cloud computing is responsible for combining and permanently storing all the data in the system to obtain a general view of the monitored environment [41,85]. In addition, the cloud accumulates some historical data over time, which can provide important information about the weather in each area [85]. We can also run machine learning algorithms on the historical data to form a smart classifier [85]. In a word, we can use fog computing to analyze a small range of data for the timely detection of disasters and warnings. Cloud computing builds long-term predictive models by analyzing data on a larger scale. In addition, cooperation between fog computing and cloud computing can improve the efficiency of data analysis through data pre-processing in fog computing and database construction in cloud computing.

Regarding data analysis methods, we can use some advanced techniques, such as machine learning [38,45,62,70,85,86,87], deep learning [46,73], and time-series data analysis [58]. Some studies may use traditional methods, such as mathematical modeling (e.g., Markov Process, Laplace Transformation) [68], observing the signal (e.g., flare marker buoys) [76], and comparing the current situation with past cases [84]. Table 2 shows data analysis methods for different types of disasters.

Machine learning techniques are commonly used to analyze the data from sensors. We can use machine learning techniques to analyze the data related to different types of disasters, such as earthquakes [38] and floods [45,62,70,85,86,87]. The machine learning techniques used in earthquake detection include convolutional neural networks [38] and recurrent neural networks [38]. They can analyze the data collected by accelerometers. The machine learning techniques used in flood detection include Bayesian Learning [45], Multi-Layer Perceptron Artificial Neural Networks [45], Random Forest [45], J 48 Decision Tree [45], Random Tree [45], Simple Cart Decision Tree [45], and BFTree [45]. They can classify and analyze the water level data. Regarding data classification, we can classify the water level data into stable level (i.e., −20°, 20°), slight increase level (i.e., 20°, 45°), high increase level (i.e., 45°, 90°), slight decrease level (i.e., (−20°, −45°) and high decrease level (i.e., −45°, −90°) [45]. Another classification is 0 to 0.25 for Mild level, 0.26 to 0.5 for the Moderate level, 0.51 to 0.75 for Severe level, and 0.76 to <1 for Critical level [87].

Furquim et al. [37] assumed that the water level data is time-series data. They modelled the time series data based on chaos theory. They used the false nearest neighbor method to estimate the value of the separation dimension and the embedding dimension. Furquim et al. [64] found that the best results for all the sensors were obtained when the separation dimension was one and the embedding dimension was two. When we adopted the distributed approach, MLP could present the peak values in a better way. The peak values are very important in examining flood prediction since they are at the points where the flooding occurs. In addition, Furquim et al. [69] used a multilayer perceptron artificial neural network to construct the recursive prediction model and obtained better results when the separation dimension was one and the embedding dimension was four. Furthermore, Chen et al. [86] suggested using a Bidirectional Gated Recurrent Unit (BiGRU) model with attention mechanisms to deal with the time-series flood data. The attention mechanism is used to automatically adjust how well the input features match the output features, while the BiGRU model aims to process the input series from both directions of the time series (chronologically and anti-chronologically) and then merge their representations together.

Since we use cameras for disaster detection more commonly, image processing algorithms have become one of the most common data analysis methods [48,71,72,73,79]. Previous studies adopted image-processing algorithms to detect floods [48,71,72,73] and typhoons [79]. Although some of them are machine learning algorithms, the data they processed was different from the machine learning mentioned above. Image processing algorithms focus on image data, while the machine learning algorithms mentioned above focus on numerical data.

After data analysis, scholars tend to present the results to the public through web applications [40,42,43,44,47,48,57,58,61,63,65,71,72,83,85,88,89]. Some of them are mobile applications such as Ferraz et al. [61]. In addition, we usually visualize data with maps to present environmental data and disaster positions in web applications [40,42,45,48,58,63,72,78,81,83,84,88]. Interestingly, Kanak et al. [81] integrated data from the sensors to create a virtual reality environment to help residents perform fire and earthquake escape drills. In addition, email [43,50,71], messages [43,71], and social media [40,68] are the ways to send out warnings to people in time. They are also the ways to send out disaster warnings.

## 5. IoT Systems in the Post-Disaster Stage

### 5.1. Sensors

Excepted for environmental data [15,90,91,92,93], the sensors in the post-disaster stage mainly collect human health data [90,92,93,94] and position data [15,38,93,94,95], which can improve the efficiency of search and rescue.

Regarding environmental data, Ochoa and Santos [15] suggested using sensors to collect environmental data in terms of weather, chemicals, and movement. Sahil and Sood [90] placed sensors on the buildings and in-pavements in the disaster-affected areas to collect environmental data, including water level, tilt in structures, temperature of buildings and ambient, smoke detection, obstacles in the path, visibility range, and location of the sensor. Korkalainen et al. [93] suggested using gas sensors to monitor air quality. Usually, multiple agencies participate in rescue operations. Each agency could use the sensors deployed in cities (e.g., weather stations, traffic cameras, wind sensors, precipitation meters, road surface condition sensors, and visibility meters) to collect environmental data [91,92]. Also, the agency integrated its own sensors, such as unmanned vehicles equipped with GPS sensors, acoustic detection, distance measurement, and motion sensing, and drones equipped with GPS sensors and cameras [91,92]. Regarding disaster mitigation, Goyal et al. [96] and Rahman et al. [47] collected water level data to open the floodgates. Rahman et al. [47] used ultrasonic sensors and a force-sensitive resistor to measure the water level. In addition, floods may cause potholes on the road. Ulil et al. [80] developed pothole monitoring systems with a modem accelerometer and gyroscope. They used the Raspberry Pi as a single computer board. They placed the sensors on the vehicles. For building structure health detection after an earthquake, Antonacci et al. [97] suggested using the LIS344ALH accelerometer, TAOS 2561 light sensor, and SHT11 temperature and humidity sensors, which were installed on the Imote2 platform and ISM400 board.

The human health data, including the rescuers’ health data and the stranded people’s health data, Boukerche et al. [94] suggested that command posts should guarantee the safety of first responders through the body-worn sensors in wearable smart devices, such as smart glasses and smart watches. Sahil and Sood [90] developed an IoT system to prioritize the evacuation of panicked, stranded people and provide them with timely medical support. They used the health sensors in stranded people’s personal mobile communication devices to collect health data, including heart rate, breath rate, dizziness, sweating, chest pain, trembling, chills, choking, nausea, and the location of the individuals. In the rescue operations suggested by Johnsen et al. [92], each rescuer was equipped with a personal sensor system (e.g., a tactical vest or tactical underwear) to monitor their health and position. The sensors in the tactical vest aimed to monitor water levels. The tactical underwear contained medical sensors, including one muscle activity sensor, pulse oximeters, and a heart rate sensor, which were installed on the Arduino single computer board. Regarding detecting epidemics in disasters, Ehsani et al. [98] used the case of detecting COVID-19 in earthquakes to explain an IoT framework. In evacuation centers and temporary hospitals, they used thermal and infrared sensors to monitor people’s body temperatures and detect fever. And they used heart rate sensors to measure oxygen levels and detect breathlessness. In affected areas, they monitored body temperature data through medical infrared thermometer guns. And they also deployed thermal sensors on unmanned aerial vehicles to monitor people’s body temperature. Korkalainen et al. [93] used CO sensors, CO_2_ sensors, optical sensors (e.g., LWIR cameras, visual range cameras), vibration sensors, sound sensors, and an ion mobility spectrometer to detect and locate life. An ion mobility spectrometer could detect volatile organic compounds, such as ammonia and Acetone [93].

The position data includes the stranded people’s position data, the rescuers’ position data, and the rescue vehicles’ position data. Firstly, people’s personal devices are effective sensors to help us collect stranded people’s position data [38]. We can ask stranded people to wear radio frequency identification bracelets to locate them as well [99]. And we can locate evacuated social vehicles through portable on-board radio frequency identification tags [99]. Suri et al. [91] also suggested using traffic cameras to search for and locate people trapped in vehicles or rubble. The traffic camera can clearly take images for every passing car, and recognize the license plate and driver characteristics automatically [99]. Furthermore, we also installed the sensors in the rescuers’ personal devices (e.g., tactical vests, tactical underwear) to locate the rescuers [92]. Ochoa and Santos [15] suggested using GPS, Radio Frequency positioning, and inertial sensors to track rescuers in the field. For rescue vehicles (e.g., ambulances, fire engines, police cars, and engineering vehicles), we should equip them with IoT equipment to locate them. For example, Anagnostopoulos et al. [95] use sensors to collect the position data of the emergency medical service system. Last but not least, Rahman et al. [47] installed GSM to locate the floodgates.

### 5.2. Communication Technologies and Protocols

Normal communication technologies and protocols are also suitable for communication in the post-disaster stage, such as Wi-Fi [38,80,93,98], Bluetooth [98], Internet [98], MQTT [47,80,92,100], LoRa/LoraWAN [92], WPAN [93], 3G/4G [80], and COAP [100]. However, disasters may destroy the infrastructure in the cities [94]. We may address this issue by maintaining the efficiency of existing communication infrastructure and using additional mobile communication tools.

To maintain communication efficiency in the face of a reduced number of communication facilities, we should develop resilient communication networks and reduce contention during data communication. Alvarez et al. [101] suggested using the Bluetooth Mesh emergency network to utilize the remaining sensors to mediate device-to-device communication in the post-disaster stage. Regarding energy-constrained IoT sensors, Ai-Turjman [102] suggested using the Cognitive Energy-Efficient Algorithm (CEEA). The CEEA was a topology-independent protocol that can handle randomness in IoT networks. The CEEA determined the path from routing nodes to sensors based on the remaining energy of each node. To be specific, the CEEA would control the remaining energy of neighbors of recent routing nodes each time before sending data from the recent routing nodes. If the energy of one of the neighboring routing nodes was less than half of its initial value, the CEEA might determine a new path to transmit the data. If the residual energy of all neighboring routing nodes are found to be below 50% of the beginning energy, the CEEA uses the same strategy. However, it was noted that multitier IoT networks and cluster- or tier-wide synchronization were the two assumptions for the effective use of the CEEA. Aljohani and Alenazi [77] introduced a multi-path resilient routing system based on software-defined networking (SDN), which combined aided-multipath routing with the capabilities of SDN. Campioni, Lenzi, Poltronieri, Pradhan, Tortonesi, Stefanelli, Suri and Ieee [100] developed a multi-domain Asynchronous Gateway of Things to enable discovery across different communication protocols and administrative domains in post-disaster relief. In addition, epidemic protocols can solve the contention caused by the reduced numbers of communication infrastructure in the post-disaster stage. Tan et al. [103] proposed an adaptive probabilistic epidemic protocol that can effectively suppresses redundant messages and reduces contention/collision levels. This protocol allowed a node to decide whether to respond to a broadcast based on information such as the number of neighbors of the broadcasting node. Ochoa and Santos [15] suggested using epidemic routing algorithms and spray and wait routing algorithms to support the dissemination of shared information among personal devices.

In addition, we can use additional mobile communication tools in post-disaster relief, such as drones [90,94,103] and vehicles participating in relief [10,22]. Tei et al. [22] proposed an opportunistic data dissemination protocol. They proposed facilitating the transmission of information from victims to rescue agencies by using the existing vehicles, including ambulances, police cars, and fire trucks, as well as the sensors inside the network. We can store the data from drones in the cloud [90]. The vehicles participating in relief include dynamic vehicles (e.g., fire trucks, ambulances) and stationary vehicles [15]. Dynamic vehicles aim to transmit data from sensors to stationary vehicles, while stationary vehicles (e.g., the base vehicles) focus on the transmission between different teams or companies in post-disaster relief [15]. Further, Johnsen, Zielinski, Wrona, Suri, Fuchs, Pradhan, Furtak, Vasilache, Pellegrini, Dyk, Marks, Krzyszton and Ieee [92] suggested using the base vehicle as a central server that utilizes Wi-Fi to receive data from sensors deployed on mobile unmanned vehicles and in cities. The video data from the drones could also be transmitted to the base vehicle through radio links [92].

### 5.3. Analysis and Applications of Sensor Data

People focus on analyzing position data and health data in order to plan evacuation routes [38,90], allocate ambulance vehicles [95], understand the health of trapped individuals [90], and integrate information for rescuers [91,92]. For example, Ehsani et al. [98] can detect COVID-19 cases in a disaster by analyzing people’s temperatures with machine learning techniques. Regarding locating people in disasters, Kristalina et al. [104] used the least squares method to improve the generalized geometric triangulation scheme, which allows sensors to track the position of rescuers or victims. Similarly, Konomi et al. [105] proposed a cooperative location inference mechanism to locate the sensors automatically. And they developed a user-participatory sensing environment that allows people to collect position data from sensors. In addition, we may carry out post-disaster activities with the cooperation of multiple agencies [9]. Thus, we need to collect data from various heterogeneous sensors. Konomi, Wakasa, Ito and Sezaki [105] proposed a novel multi-factor cost model to integrate the multi-modal sensor data consistently and flexibly. Li et al. [106] introduced a semi-automated role mapping process for dynamic cross-domain accesses of sensors in post-disaster relief to solve heterogeneity and protect sensitive information.

In addition, the combination of fog computing and cloud computing can improve the efficiency of data analysis [90,91,92]. For example, Suri et al. [91] proposed the Sieve, Process, and Forward (SPF) Fog-as-a-Service platform to address the scenario of post-disaster relief. Fog computing is also helpful in reducing the time of search and rescue since it can improve the efficiency of data analysis through pre-processing some data analysis, such as data categorization and data novelty analysis [90,91]. Cloud computing aims to store the data and process deeper data analysis [90,91,92,98]. The fog layer exists in the gateway (e.g., drones and evacuation vehicles) and serves as a bridge between the sensor layer and cloud layer of the Internet of Things [90,91]. The utilization of the fog layer is attributed to its position awareness and close proximity to the sensors [90]. This enables it to perform essential data pre-processing tasks, such as data categorization, novelty analysis, panic health status classification, and alarm creation [90]. Due to the inherent limitations in computing and storage capacities of the fog layer, the cloud layer was employed to store and analyze environmental and health data, as well as the corresponding panic health status data [90]. This facilitated the generation of alerts in the form of compiled medical records [90]. The Cloud layer includes temporal data mining, cloud storage, panic health sensitivity monitoring, evacuation strategy, and evacuation map building [90]. Thus, we usually develop an application to combine the data and conduct big data analysis on the cloud [91,92]. And we will place the mobile decision-making centers for post-disaster rescue on the cloud [91,92].

Regarding data application, scholars usually publish the results of data analysis on the web application [15,38,88,91,92,99,105]. Some of them are mobile applications [15,92,105]. We visualize data with maps to present the collapse, the traffic, the shelters, and the evacuation route in the post-disaster stage [15,38,88,90,91,92]. For example, Lwin et al. [88]’s City Geospatial Dashboard can provide road congestion information to help disaster response teams estimate travel times to reach disaster areas. toRoute planning is one of the major applications in the post-disaster stage [38]. Based on position data, Kim et al. [38]’s system can provide a real-time evacuation route guidance service, including searching for the safest shelter and showing pedestrian paths for users. Then, previous studies tend to use dynamic programming methods to support route planning. Konomi et al. [105] developed a user sensing environment to collect geo-tagged sensor data, omnidirectional cameras, and environmental sensors (e.g., temperature, humidity) to solve dynamic routing planning problem in the post-disaster stage. Liu and Wang [99] suggested using the variable structure discrete dynamic Bayesian network model for real-time dynamic path planning This approach could facilitate the prompt evacuation of social vehicles from the event area, while also ensuring the timely arrival of rescue vehicles at their allocated position to carry out necessary tasks. Anagnostopoulos et al. [95] developed a real-time dynamic routing algorithm for the ambulance arrangement to reduce the time per route, distance covered, and fuel consumption.

Some scholars designed mitigation measures specifically for floods. Goyal et al. [96] use reinforcement learning to develop gate-control systems on Flash Flood Bypass Waterways to evacuate the flood water to channels. Rahman et al. [47] also controlled the valves based on water level data collected by sensors. In addition, Ulil et al. [80] developed pothole monitoring systems to detect potholes on the road caused by floods. Decision trees and machine support vector methods are the data analysis within the systems.

## 6. Comparison between the IoT Applied in Pre-Disaster and Post-Disaster Stages

We usually use sensors to collect environmental data in the pre-disaster stage. However, we should use sensors to collect health data and position data more in the post-disaster stage. We can transmit the data with a number of communication technologies and protocols in the pre-disaster stage. However, disasters may destroy communication infrastructures. Thus, it is important to maintain the efficiency of existing communication infrastructure and use additional mobile communication tools in the post-disaster stage. Machine learning techniques are common data analysis methods both in the pre-disaster stage and the post-disaster stage. With the development of image processing algorithms, we used cameras as sensors more. Fog-cloud computing is useful to improve the efficiency of data analysis both in the pre-disaster and post-disaster stages. The data applications aim to provide environmental information and send warnings in the pre-disaster stage. However, they focus more on route planning in the post-disaster stage.

In addition, five publications mention IoT systems used in both the pre-disaster and post-disaster stages [38,47,88,94,105]. Environmental data should be collected both in the pre-disaster and post-disaster stages since we need to use environmental data to detect disasters in the pre-disaster stage and ensure the safety of the environment in the post-disaster stage. For example, Rahman et al. [47] may use water level data to control the valves to prevent sewerage system overflow and mitigate floods. Since position data is also useful in the post-disaster stage, people can check environmental data and position data in Lwin et al. [88]’s application in the pre-disaster and post-disaster stages. Furthermore, Kim et al. [38] and Konomi et al. [105] provide evacuation route planning in the applications.

## 7. Conclusions and Future Work

Disasters have the potential to inflict harm upon human lives and result in significant economic and environmental repercussions, particularly in densely populated urban areas [6,7]. Given the substantial magnitude of losses incurred by urban disasters, it is imperative to enhance the efficacy of urban disaster management. Furthermore, the implementation of effective disaster management strategies is of utmost importance for urban areas to successfully attain the UNSDGs [12]. Sensors play a crucial role in the acquisition of data within IoT devices. They serve as a connection between the physical and digital realms, playing a vital function within the IoT framework. IoT has the potential to offer real-time monitoring, early warning systems, post-disaster response, and rescue support, thereby playing a significant role in the field of urban catastrophe management.

This study conducted a SLR to examine the utilization of sensors in the IoT for urban catastrophe management. The evaluation encompassed both the pre-disaster and post-disaster stages and analyzed a total of 72 publications. This study fills the research gap in sensors on IoT systems for urban disaster management. Nevertheless, this study possesses certain limitations that warrant further investigation in order to enhance its findings. The utilization of keywords has the potential to restrict the size of the sample. The formulation of keywords and establishment of inclusion criteria are based on the research questions, which may lead to the exclusion of some articles. Additionally, the selected publications may place a greater emphasis on the topics of earthquakes, landslides, and floods. There are a limited number of scholarly works that specifically address the topic of natural disasters, such as storms.

In addition, we explore potential avenues for future study in the field of disaster management. Specifically, we highlight the following areas: sensor heterogeneity, post-disaster emergency communication, integration of sensor technology with unmanned aerial vehicles, user participation in sensing, and the calculation of post-disaster rescue time.

Firstly, it can be observed that a diverse range of sensors are employed in the field of disaster management, encompassing both the pre-disaster and post-disaster stages. The presence of various manufacturers and the diverse applications of sensors in disaster scenarios contribute to the heterogeneity of these sensors, hence hindering the integration and sharing of information [107,108]. Some disasters may cause sequent disasters. For example, seismic activity or inundations can cause floods [109]. By performing an analysis of sensor data pertaining to various sorts of disasters, it becomes possible to anticipate the occurrence of subsequent disasters following an initial one. Moreover, future research endeavors could explore the integration of disparate sensors in order to develop a holistic application capable of facilitating the visualization of sensor data and the dissemination of alerts pertaining to various categories of disasters. To facilitate communication among heterogeneous sensors, future research endeavors may explore novel communication technologies and protocols, including the incorporation of integration brokerage applications. The JosNet system serves as a brokerage platform that facilitates interoperability and integration among many low-rate and low-power protocols, including Bluetooth LE, Zigbee, and Thread [110]. Furthermore, integration brokerage programs provide seamless communication with the remaining sensors and other heterogeneous devices throughout the post-disaster phase, hence enabling sustained communication in this critical time.

Secondly, in the aftermath of a disaster, it is possible to employ low-power communication technologies and protocols to sustain communication. For instance, one such technology is low-power satellite communication protocols. Random-access, very-low-power, and wide-area networks (RA-vLPWANs), as low-power satellite communication protocols, provide uncoordinated multiple access in scenarios characterized by poor signal-to-noise ratios and very low signal power [111]. Furthermore, the CEEA algorithm, as suggested by Ai-Turjman [102], serves the purpose of establishing a post-disaster sensor network by effectively connecting operational sensors with remaining energy. Notably, this algorithm bears a resemblance to the Bee algorithm [112]. Future research endeavors may be directed toward examining the potential applicability of the Bee algorithm in optimizing sensor networks during the post-disaster phase.

Thirdly, future studies also can combine sensors with unmanned aerial vehicles. We can install sensors on unmanned aerial vehicles. Future studies may investigate what types of sensors are suitable to install on unmanned aerial vehicles. If we use cameras on unmanned aerial vehicles, it is useful to investigate how to transmit the image data from unmanned aerial vehicles to the operation centers. Since we may control unmanned aerial vehicles remotely, how do we ensure the connects between controllers and unmanned aerial vehicles? Unmanned aerial vehicles can also serve as communication tools, especially in the post-disaster stage. It is interesting to optimize unmanned aerial vehicles’ cruising trajectory to balance the communication coverage and cost. Future studies may also investigate how to use unmanned aerial vehicles to locate the sensors or trapped people.

Fourthly, some studies mentioned user participation in sensing in the pre-disaster stage. However, people may upload false information to the system, affecting the credibility of the system. Future studies may investigate how to ensure the authenticity of data with Blockchain. In addition, because of the wide application of cameras, sensor communication must be able to transmit a larger amount of image data.

Fifthly, future studies can use sensors to calculate the time from departure to the successful completion of the rescuers in the post-disaster stage, which can allocate rescue personnel more efficiently. For example, personal devices can record the rescuers’ routes and calculate the time. The rescuers may record the completion of the rescue when they save the people successfully.

## Figures and Tables

**Figure 1 sensors-23-07475-f001:**
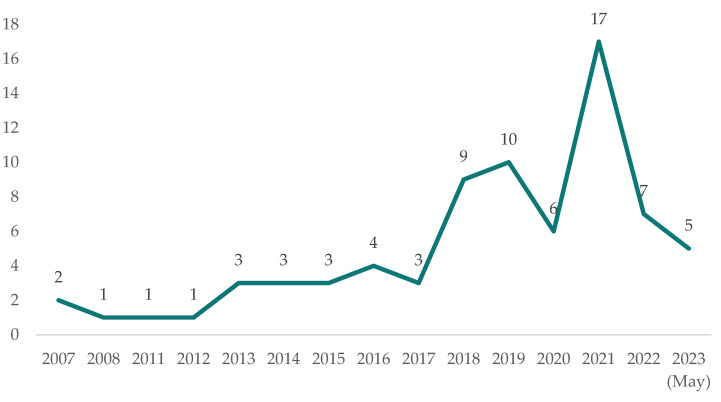
The distribution of publications by year.

**Table 1 sensors-23-07475-t001:** Top six distribution of publication by sources.

Source	Number of Publications
Sensors	5
2019 5th IEEE International Smart Cities Conference (IEEE ISC2 2019)	2
Applied Sciences	2
Earth Science Informatics	2
IEEE Access	2
Materials Today-Proceedings	2

**Table 2 sensors-23-07475-t002:** Data analysis method used in analyzing data from sensors.

Types of Disasters	Method	Detail	Reference
Earthquake	Machine learning	Convolutional Neural Network	Kim et al. [38]
		Recurrent Neural Network	
Landslide	Time-series data analysis	Grey System Forecasting	Miao and Yuan [58]
Floods	Machine learning	Bayesian Learning	Furquim et al. [45]
		Multi-Layer Perceptron Artificial Neural Networks	
		Random Forest	
		J-48 Decision Tree	
		Random Tree	
		Simple Cart Decision Tree	
		BFTree	
Floods	Machine learning	Artificial Neural Networks	Wang and Abdelrahman [62]
		LSTM	
Floods	Machine learning	Random Forest	Aljohani et al. [85]
		Decision Tree	
		KNN	
Floods	Machine learning	Artificial Neural Network	Mousa et al. [70]
Floods	Machine learning	Artificial Neural Network	Goyal et al. [87]
Floods	Deep learning	Deep Neural Network	Junior et al. [73]
Floods	Time-series data analysis + Machine learning	BiGRU Neural Network + Attention Mechanism	Chen et al. [86]
Floods	Time-series data analysis + Machine learning	Multilayer Perceptron artificial neural network	Furquim et al. [37]
Floods	Time-series data analysis + Machine learning	Multilayer Perceptron artificial neural network	Furquim et al. [64]
Floods	Time-series data analysis + machine learning	Multilayer Perceptron artificial neural network	Furquim et al. [69]
Floods	Image processing algorithm		Garcia et al. [48]
Floods	Image processing algorithms	Edge Keeping Index	Liu et al. [71]
		SURF.	
Floods	Image processing algorithm	Color segmentation	Castro et al. [72]
		Morphological operations	
		Shape detection	
Floods	Image processing algorithm	DNN Pruning Algorithm + Randomized Heuristic	Junior et al. [73]
Floods	Mathematical modeling	Markov Process	Tyagi et al. [68]
		Laplace Transformation	
Floods	Data retrieval	Compare current situations with past cases	Luo et al. [84]
Typhoon	Image processing algorithm	Attention Mechanism	Wang et al. [79]
		Fast R-CNN	
		Transfer Learning method	
Tsunami	Observation	Flare marker buoys	Alhamidi et al. [76]

## Data Availability

Not applicable.

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
