# Peer review of "Sensors on the Internet of Things Systems for Urban Disaster Management: A Systematic Literature Review"

_sensors, 2023, doi:10.3390/s23177475_

Round 1
Reviewer 1 Report
The content of this paper was a review of the IOT systems for the urban disaster management system. As the requirement of a review paper, the content of this paper is too poor to be recognized as a review paper.
As the claim by the authors, the number of useful papers was limited.
“Among 19 publications, 7 publications are full research articles while 12 publications 122 are conference papers (Figure 2).”
The content of this paper only has the description of the previous paper. Its lacks of analysis, comparison, and evaluation of the problems of the application of the IOT on the subject. The authors should make reference to the other review papers that have been published in Sensors.
The style and grammar of English need to be improved. Many mistakes are found in this manuscript.
Extensive editing of English language required
Author Response
Thank you for your kind comments. We reviewed more 37 papers in our paper. Please read the analysis based on new papers on the line 120-476. We will process English editing then. If there are more issues, please feel free to give further comments. We will spare no effort to revise the manuscript accordingly. Thank you.
Reviewer 2 Report
The paper discusses an interesting topic. However, the paper needs the following major modifications:
1- Language recheck : grammar and meanings
2- The abstract does not show why such literature review is needed and its importance. Also, no references should be in the abstract.
3- restructuring the introduction section to: general overview, previous systematic reviews in the same field, contribution and the importance of the proposed review
4- More references should be used.
5- why the review starts with 2017 ,, we believe there are many disaster management methods were proposed before that
6- avoid the word "results" in the title of sections 4 and 5. Results indicate the findings ... We suggest make descriptive title -e.g. IoT systems in pre-disaster stage
7- in each subsection 4.1 and 5.1 comparison is needed between types of sensors, it is better to use table .. similarly 4.2 and 5.2 between types of communications.
8- a new section should be added to highlight the differences in technologies between pre-disaster and post-disaster . for example, what are the differences in sensors, communications and data analysis between pre and post disaster cases. how emergency affects the use of IoT technologies
9- a conclusion section is needed to summarize the work, show the limitations and suggest future work
10- such review can be more useful to the researchers in the field if it is made as a taxonomy
11- each category can be made clearer if an architecture is provided. Adding more illustrative figures to show the main ideas and differences
1- Language recheck : grammar and meanings
Author Response
Thank you for your kind comments. We use 37 more references. We also highlight the differences in technologies between pre-disaster and post-disaster. And we intergate the idea in each reference. Please read the analysis based on new papers on the line 120-476. After adding more references, we find that the review starts with 2007. We remvoed the word "results" in the title of sections 4 and 5. And we changed the title to "IoT systems in the pre-disaster stage" (line 131) and "IoT systems in the post-disaster stage" (line 362). If there are more issues, please feel free to give further comments. We will spare no effort to revise the manuscript accordingly. Thank you.
Reviewer 3 Report
The main contributions and motivations for the present work have to be clearer.
Authors need to confirm that all acronyms are defined before being used
The authors’ examination of the manuscript was not careful enough. For example, when introducing some English abbreviations, their full names are not capitalized,
The contributions of this manuscript need further refinement, and some of them seem insignificant. please describe the features, functions and innovations of the network in more detail.
In introduction the authors can come up with the existing survey works on the similar topic, probably summary table.
The research method is not clear, please clarify the research method involved.
Why the authors have chosen only this model why not others must be more clear.
Author Response
Thank you for your kind comments. We provided more details of the features, functions and innovations of the IoT systems used in pre-disaster and post-disaster stages. Please kindly find in line 131-476. If there are more issues, please feel free to give further comments. We will spare no effort to revise the manuscript accordingly. Thank you.
Round 2
Reviewer 1 Report
The revised version has been improved significantly. However, it lacks commendation and suggestions for the “Sensors on the Internet of Things Systems for Urban Disaster Management”.
For a review paper, please provide your commendation and suggestions.
Minor editing of English language required
Author Response
Thank you for your kind comments. To provide our commendation and suggestions, we revise section 4 and section 5 (Line 167-592). For example, we develop a table to show the data analysis method and applications. Please kindly find in the line 367 and 393 on the page 8-10. And we develop a new section to compare the IoT systems in pre-disaster stage and post-disaster stage. Please kindly find in the line 593-603 on page 15. In addition, we revise the introduction to better explain why we develop this review. Please kindly find in the line 24-84 on page 1-2. The final number of sample pulications has increased to 73. If there are some problems, please do not hesitate to point out. Thank you.If there are some problems, please do not hesitate to point out. Thank you.
Reviewer 2 Report
The authors still don't answer my comments.
A revised version should contain the following , or the authors have to justify why not
. Why such review is needed to be published
. What is the difference between predisaster and post disaster technologies,,sensors,, communications and data analysis .. a table should be added to show the difference
. A conclusion or summary is needed
English is improved after the first round.. minor check is needed
Author Response
Thank you for your kind comments. We revise the introduction to answwer the first question "why such review is needed to be published". Please kindly find in the line 24-84 on page 1-2. We develop section 6 to answer the question "What is the difference between predisaster and post disaster technologies,,sensors,, communications and data analysis .. a table should be added to show the difference". Please kindly find in the line 593-603 on page 14-15. And we develop the conclusion in the line 604-621 on page 15. In addition, we revise section 4 and section 5. Please kindly find in the line 167-592 on page 5-14. The final number of sample pulications has increased to 73. If there are some problems, please do not hesitate to point out. Thank you.
Round 3
Reviewer 1 Report
The content of the revised version has improved significantly
Minor editing of English language required
Author Response
Thank you for your kind comments. We also revised the manuscript this time. We organize the section 4 and section 5 again to make it more flow. And we conduct proofreading to revise the grammaar errors on the manuscript.
Reviewer 2 Report
- The authors still need to make some minor revisions:
1- English check, there are many mistakes
2- Table 2, the method column, why some disasters have method and some do not have ,,, For example row 6,, this should be deep learning
3- add reference to section 6. e.g. cite papers about using drones as a base station...reference for combining edge with cloud in
Edge–Fog–Cloud Computing Hierarchy for Improving Performance and Security of NB-IoT-Based Health Monitoring Systems
Iot-based system for improving vehicular safety by continuous traffic violation monitoring
minor English revision
Author Response
1- English check, there are many mistakes
→Thank you for your kind comments. We conducted proofreading for the manuscript. We revised the grammar errors on the manuscript.
2- Table 2, the method column, why some disasters have method and some do not have ,,, For example row 6,, this should be deep learning
→Thank you for your kind comments. All of them have method on table 2 (line 425). And we change row 6 to “deep learning”.
In addition, we organize the section 4 and section 5 again to make it more flow. And we elaborated the section 6 and section 7.